# Incretin Hormone Secretion in Women with Polycystic Ovary Syndrome: Roles of Obesity, Insulin Sensitivity and Treatment with Metformin and GLP-1s

**DOI:** 10.3390/biomedicines12030653

**Published:** 2024-03-14

**Authors:** Andrea Etrusco, Mislav Mikuš, Antonio D’Amato, Fabio Barra, Petar Planinić, Trpimir Goluža, Giovanni Buzzaccarini, Jelena Marušić, Mara Tešanović, Antonio Simone Laganà

**Affiliations:** 1Unit of Obstetrics and Gynecology, “Paolo Giaccone” Hospital, Department of Health Promotion, Mother and Child Care, Internal Medicine and Medical Specialties (PROMISE), University of Palermo, 90127 Palermo, Italy; etruscoandrea@gmail.com (A.E.); antoniosimone.lagana@unipa.it (A.S.L.); 2Department of Obstetrics and Gynecology, Clinical Hospital Center Zagreb, 10 000 Zagreb, Croatia; pplaninic@gmail.com (P.P.); goluzat@gmail.com (T.G.); 3Unit of Obstetrics and Gynecology, Department of Interdisciplinary Medicine (DIM), University of Bari “Aldo Moro”, Policlinico of Bari, Piazza Giulio Cesare 11, 70124 Bari, Italy; antoniodamato19@libero.it; 4Unit of Obstetrics and Gynecology, P.O. “Ospedale del Tigullio”—ASL4, 16043 Chiavari, Italy; fabio.barra@icloud.com; 5Department of Health Sciences (DISSAL), University of Genoa, 16132 Genoa, Italy; 6Department of Obstetrics and Gynaecology, IRCCS San Raffaele Scientific Institute, Vita-Salute San Raffaele University, 20132 Milan, Italy; giovanni.buzzaccarini@gmail.com; 7University Department of Health Studies, University of Split, R. Boskovica 35, 21 000 Split, Croatia; jelena.marusic@hormona.hr; 8Hormona Polyclinic, Kranjceviceva 45, 21 000 Split, Croatia; 9Department of Obstetrics and Gynecology, General Hospital Dubrovnik, 20 000 Dubrovnik, Croatia; mara.tesanovic56@gmail.com

**Keywords:** polycystic ovary syndrome, metformin, obesity, GLP-1 receptor agonists, insulin sensitivity

## Abstract

Background: The purpose of this narrative review is to describe the mechanisms that are responsible for the development of infertility and PCOS, with a focus on the role of obesity, insulin sensitivity and treatment with metformin and GLP-1s. Methods: The relevant publications were identified after systematic queries of the following sources: PubMed, Google Scholar, Web of Science, and publishers’ databases, complemented by a cross-check of the reference lists. We used a combination of the search terms “polycystic ovary syndrome”, “obesity” and “insulin resistance” with “metformin”, “exenatide”, “liraglutide”, “semaglutide”, “orlistat” and terms relevant to the topic of each paragraph (e.g., “pathophysiology”, “metabolism”, “infertility”, “treatment”). Results: All articles describing the mechanisms responsible for the development of infertility and PCOS, with a focus on the role of obesity, insulin sensitivity and treatment with metformin and GLP-1s, were considered for this review. Conclusions: The existing research on GLP-1 receptor agonists (GLP-1RAs) has not conclusively established a specific therapeutic use for these drugs. Additionally, the efficacy of the newer generation of GLP-1RAs, particularly in terms of dosage and duration of exposure, warrants more extensive research. Understanding the optimal dosing and treatment duration could significantly enhance the therapeutic use of GLP-1RAs in managing PCOS and its associated conditions.

## 1. Introduction

Obesity is a clinical, social and economic problem that affects all states globally, from the most developed to the developing world [1]. Year after year, there has been a dramatic increase in the number of people who are overweight or obese. In 2016, there were nearly 2 billion overweight individuals, of whom about 700 million were obese [2]. Weight gain shortens the life expectancy of the general population, increasing the incidence of cardiovascular disease, diabetes, respiratory disease and cancer [3,4,5,6,7].

Women of reproductive age are not spared from this dramatic trend; in the United States, about 40% of women at this age of life are obese [8]. The hyperestrogenic status that is typical of obesity increases the risk of pre-cancerous lesions and endometrial malignant neoplasms [9]. A high body weight caused by excess fatty tissue in the abdominal cavity, which increases intra-abdominal pressure and consequently also intra-vesical pressure, is involved in the genesis of urogynecological problems such as overactive bladder [10]. In addition, obesity is responsible for reduced reproductive potential due to its negative influence on the hypothalamic–pituitary–ovarian (HPO) axis, the reduction in oocyte quality and alterations in endometrial receptivity [11,12,13].

Regardless of how pregnancy is sought, spontaneously or through assisted reproductive techniques (ART), obese women show worse reproductive outcomes than average-weight women [14,15].

Polycystic ovary syndrome (PCOS) is the most common endocrine disorder among young women of reproductive age and one of the most common causes of subfertility or infertility [16]. The diagnosis and treatment of polycystic ovary syndrome (PCOS) have been controversial, with difficulties in defining individual components within the diagnostic criteria and significant clinical heterogeneity that is further influenced by ethnic differences and changes in clinical features throughout life. Today, the diagnosis of PCOS is made according to the criteria proposed in 2003 by the Rotterdam ESHRE/ASRM PCOS Consensus Workshop Group, which defines PCOS as the presence of at least two of the following criteria: oligomenorrhea and/or anovulation, clinical and/or biochemical signs of hyperandrogenism, and polytheistic echo-structure of the ovary [17]. However, many women with PCOS are also obese, and obesity significantly influences the symptoms, management and reproductive outcomes of these patients [18].

Although healthy lifestyle intervention plays a key role in the prevention and management of excess weight in PCOS and obese women in general [19], the role of pharmacological hypoglycemic and anti-obesity agents in achieving and maintaining weight loss and providing potential health benefits is increasingly recognized [20]. Because many of these patients present serious difficulties in adhering to dietary and exercise plans aimed at improving their lifestyle and reducing their body weight, the addition of these agents seems reasonable to ensure the achievement of therapeutic goals for patients with obesity [21].

Despite the variety of distribution of these drugs, which varies from country to country, and the cost that to date often remains prohibitive, there is increasing entry of these drugs into clinical practice [22].

However, in PCOS and in women of reproductive age in general, the role of anti-obesity pharmacologic agents remains unclear, and in the latest ESHRE guidelines on PCOS, only agents approved by multiple regulatory agencies for weight management have been the subject of recommendations, including exenatide, liraglutide, semaglutide and orlistat [21].

The purpose of this narrative review is to describe the mechanisms responsible for the development of infertility and PCOS, with a focus on the role of obesity, insulin sensitivity and treatment with metformin and GLP-1s.

## 2. Materials and Methods

We adhered to the quality standards for narrative reviews, as defined and quantified by “SANRA—a scale for the quality assessment of narrative review articles” [22]. The relevant publications were identified after systematic queries of the following sources: PubMed, Google Scholar, Web of Science and publishers’ databases, complemented by a cross-check of the reference lists. We used a combination of the search terms “polycystic ovary syndrome”, “obesity” and “insulin resistance” with “metformin”, “exenatide”, “liraglutide”, “semaglutide”, “orlistat” and terms relevant to the topic of each paragraph (e.g., “pathophysiology”, “metabolism”, “infertility”, “treatment”). We did not apply any language restrictions.

This study aimed to ask the following PICO questions:

Population: women with verified PCOS according to the relevant criteria.

Intervention: assessment of metformin and GLP1 receptor agonists.

Comparison: metformin and GLP1 receptor agonist comparisons and their role in insulin resistance and obesity.

Outcome: therapeutic efficacy of metformin and GLP1 receptor agonists.

## 3. Results

All articles describing the mechanisms responsible for the development of infertility and PCOS, with a focus on the role of obesity, insulin sensitivity and treatment with metformin and GLP-1s were considered for this review. Only original papers that reported specific experience data on the topic were included. Relevant aspects of every article were recorded and commented, with particular attention to the pathophysiological considerations, metabolic and fertility consequences and current treatment options. We included 17 articles in this review, while the remaining selected articles were used for a better understanding of the role of incretin hormone secretion in women with PCOS (Figure 1).

### 3.1. PCOS: Definition and Current Pathophysiological Considerations

PCOS is a heterogeneous and complex endocrine disorder affecting up to 15% of reproductive-age women, depending on the studied population and used criteria. It can present with mild signs and symptoms in some patients, while others might develop severe disruptions in metabolic, endocrine and reproductive function [23]. The main factors linked to the pathophysiology of PCOS are ovulatory dysfunction, insulin resistance, hyperandrogenism, and inadequate pulsation of gonadotropin-releasing hormone (GnRH) accompanied by aberrant gonadotropin production. These elements of the PCOS pathophysiological cycle interact and precipitate one another, potentially leading to significant disturbances in various functions of organ systems.

Androgen hypersecretion, which is related to abnormal follicular growth and ovulatory disruption, is the main characteristic of ovary dysfunction in PCOS. Aberrant development of follicles leads to the polycystic morphology of ovaries and a build-up of anti-Müllerian hormone (AMH), resulting in modification of the ovarian microenvironment and further worsening of its function [24,25]. Moreover, hyperandrogenism creates an improper pulsatile secretion of GnRH, resulting in inappropriate gonadotropin production and excess secretion of luteinizing hormone (LH). LH has a stimulatory effect on theca cells, causing the hyperproduction of androgens, while follicle-stimulating hormone (FSH) stimulates granulosa cells, leading to increased production of estrogens [26,27].

PCOS is a polygenic and multifactorial disorder. Although many genes have been associated with this syndrome, the underlying signaling pathways and molecular components involved in its pathogenesis are still not fully understood [28]. It is currently believed that the exposure of individuals who exhibit predisposing genetic characteristics to some of the significant environmental triggers leads to the development of PCOS phenotypes [29,30]. According to the evidence from newer studies, there is a potential environmental risk during both the prenatal and postnatal periods. Intrauterine exposure to high concentrations of circulating androgens or glucocorticoids in mothers with PCOS during critical stages of fetal development may determine the phenotypic expression of PCOS in adulthood [31,32]. Postnatal environmental factors include lifestyle factors such as diet and nutrition, environmental toxins and socioeconomic status [33,34]. Furthermore, recent research has shown that the gut microbiomes of adult patients with PCOS and several PCOS models are different from those of healthy adults and control animals, suggesting that gut microbiota may also play a significant role in the pathogenesis of the disease [35,36,37].

PCOS was first described in 1935 and today is defined according to several criteria: the 1990 National Institutes of Health criteria, the 2003 Rotterdam criteria, the 2006 Androgen Excess and PCOS Society criteria, and the 2007 Japan Society of Obstetrics and Gynecology criteria [38,39,40]. The National Institutes of Health consensus conference panel recommended further specifying the phenotypes of PCOS in phenotypes A, B, C, or D (Table 1).

Commonly, individuals with PCOS display both hyperandrogenism and inconsistent menstrual cycles, which generally makes further evaluation for PCOM unnecessary. In the case of adolescents, diagnosis necessitates the presence of both ovulatory issues and hyperandrogenism [21].

Indications of ovulatory dysfunction are often identified by the irregularity of menstrual cycles, specifically cycles lasting fewer than 21 days or more than 35 days, especially if occurring over three years post-menarche. According to the evidence-based guidelines (EBG), fluctuations in cycle regularity are typical in the initial year following menarche [21].

As the primary clinical manifestation of hyperandrogenism is hirsutism, biochemical hyperandrogenism is gauged by measuring the levels of total or free testosterone, or by computing the free androgen index. Liquid chromatography–mass spectrometry (LC-MS) is the preferred method for determining testosterone levels, as direct immunoassays are not as sensitive or accurate, especially at lower hormone concentrations. Despite the widespread availability of LC-MS, many laboratories in Nordic regions continue to use direct immunoassays, which require vigilance from healthcare professionals. PCOM is determined by observing more than 20 follicles, each measuring 2–9 mm, in either ovary. Additionally, adult individuals can be evaluated for PCOS using AMH levels, although these are influenced by variables such as age, BMI, ethnicity, contraceptive use, and menstrual cycle phase. The appropriate AMH thresholds are dependent on the population being studied and the specific assays used. It is crucial to note that neither PCOM nor AMH should be utilized for diagnosis within eight years following menarche due to the naturally high ovarian reserve in adolescents [21].

In addition to this, a recent study found that in women with PCOS and infertility, higher AMH concentrations were linked to reduced chances of ovulation when treated with OI using clomiphene, clomiphene combined with metformin, or metformin alone. For this reason, AMH levels in women with well-characterized PCOS could be positively implemented as diagnostic criteria for PCOS in the near future [41].

Recently, a metanalysis on 24 studies that comprised 20 adult studies (with 3883 controls and 3859 PCOS individuals) and four adolescent studies (with 210 controls and 268 girls with PCOS) evaluated the diagnostic accuracy of various ultrasonographic features of ovarian morphology in diagnosing PCOS. Indeed, there is still uncertainty about the optimal ultrasonographic markers used to define polycystic ovary morphology (PCOM). For adult women, follicle number per ovary (FNPO) emerged as the most accurate diagnostic marker (sensitivity: 84%, specificity: 91%; AUC: 0.905). Ovarian volume (OV) and follicle number per single cross-section (FNPS) showed similar sensitivities but lower specificities and AUCs compared to FNPO. In adolescents, OV appeared as a promising marker, though this evidence is still limited. For this reason, FNPO emerged as the gold standard in ultrasonographic diagnosis of PCOS in adults, with OV and FNPS used as alternatives [42].

### 3.2. Metabolic and Fertility Consequences of PCOS

As a complex and multifaceted disorder, PCOS is linked with many long-term metabolic consequences. Women with PCOS have an increased risk of developing glucose intolerance and diabetes mellitus, dyslipidemia, chronic hypertension, systemic inflammation, and coagulation disorders [43]. All these factors may further increase the risk of developing cardiovascular disease (CVD) and CVD-associated morbidity and mortality. It has been shown that women with PCOS have a greater prevalence of abnormal lipid profiles, including increased low-density lipoprotein (LDL) cholesterol and very-low-density lipoprotein (VLDL) cholesterol levels, high serum triglyceride and free fatty acid concentrations, as well as decreased high-density lipoprotein (HDL) cholesterol levels, which are attributed to decreased apolipoprotein A-I (apoA-I) levels [44,45,46]. Due to the smaller and dense particles of LDL cholesterol, a lipid profile in patients with PCOS is more atherogenic, which is further aggravated by the presence of insulin resistance and obesity [47]. The strong link between obesity and PCOS is substantiated by epidemiological data, which show that 38–88% of women with PCOS are overweight (BMI 25–29.9 kg/m^2^) or obese (BMI >30 kg/m^2^) [48,49,50]. In both lean and obese women with PCOS, adipose fat is distributed mostly in the android pattern—predominantly in the visceral and upper thoracic parts of the body [51,52,53]. Abdominal obesity has been identified as a significant risk factor for developing CVD, as well as a high waist-to-height ratio (an index of abdominal obesity; also often present in patients with PCOS), which is shown to be strongly linked to diabetes mellitus type 2 and metabolic syndrome [54,55]. Visceral adipose tissue secretes many molecules that can directly affect adrenal and ovarian function, consequently affecting the metabolism of steroid hormones. Adipokines such as leptin and proinflammatory cytokines such as TNFα, IL-6 and IL-18 are involved in the pathogenesis of obesity-related insulin resistance in women with PCOS but seem also to be directly implicated in the pathogenesis of ovarian and adrenal dysfunction [56,57,58]. Increased levels of C-reactive protein (CRP), an indicator of systemic inflammation, have additionally been found in patients with PCOS, further contributing to an increased risk of the development of cardiovascular diseases [59,60].

Women suffering from PCOS usually also experience reproductive dysfunction, including both subfertility and early pregnancy loss. Furthermore, there is a higher risk of premature birth after either spontaneous or assisted conception in these women [61]. Dysregulated folliculogenesis and abnormal steroidogenesis are the main characteristics of ovarian dysfunction in infertile patients with PCOS. There are several aberrant physiological pathways involved in the pathogenesis of chronic anovulation, and many of them are still being studied [62] (Table 2). Potential etiological factors included in early pregnancy loss in women with PCOS other than the already mentioned hypersecretion of luteinizing hormone, hyperinsulinemia and hyperandrogenaemia include altered endometrial receptivity, abnormalities in plasminogen activator inhibitor (PAI) activity and abnormal vascular responses due to endothelial dysfunction [63,64]. The most recent publications have revealed the molecular mechanisms that are responsible for reproductive issues in this particular population. For instance, a study by Jiang and associates demonstrated hyperinsulinemia as the main factor in PI3K/AKT-NR4A pathway disruption which leads to decidualization defects and endometrial dysfunction, thereby contributing to infertility in PCOS patients [65]. All these possible factors are interlinked and potentially exacerbated by each other, resulting in fertility issues and other reproductive challenges for women with this disease.

### 3.3. Specific Role of Obesity and Insulin Resistance in PCOS

Among the myriad factors contributing to the development and manifestation of PCOS, obesity and insulin resistance (IR) have emerged as key players. The intricate interplay between these two factors and PCOS is multifaceted, influencing both the pathogenesis and clinical outcomes of the syndrome. Obesity, characterized by an excess accumulation of adipose tissue, is closely linked to PCOS, contributing to the severity of symptoms and exacerbating hormonal imbalances. The impairment of lipid metabolism has systemic consequences on the human body. As evidence, PCOS has been recently included among the risk factors for cardiovascular disease (CVD) in the latest 2023 PCOS ESHRE guidelines [21]. Women with PCOS display an higher prevalence of obesity [65,66]. Furthermore, women with a BMI >25 kg/m2 face less retrieved oocytes in IVF cycles, have a lower chance of pregnancy following IVF, require higher doses of gonadotrophins and have increased miscarriage rates [67,68]. Additionally, IR plays a crucial role in the metabolic disturbances observed in PCOS. Women diagnosed with PCOS often manifest insulin resistance (IR) and hyperinsulinemia, independent of their adiposity level or androgen levels [69]. Moreover, females diagnosed with PCOS face an elevated likelihood of developing type 2 diabetes (T2D) and experiencing impaired glucose tolerance (IGT) [70]. This interconnection is a critical aspect of the syndrome, as IR not only contributes to hyperinsulinemia but also has a direct impact on ovarian function, aggravating the reproductive abnormalities associated with PCOS. The pathophysiological heterogeneity of PCOS is reflected by the variable prevalence rates of IR in PCOS-affected women reported in the literature, ranging from 44% to 70% in the majority of studies [71].

Insulin receptors can be identified in the theca and granulosa cells, playing a role in coordinating metabolic, steroidogenic and mitogenic functions [72]. In human theca cells, insulin has been demonstrated to directly enhance the activity of 17-alpha hydroxylase (CYP17A1) through the PI3K pathway, operating independently of MAPK [73]. Insulin receptor substrate-1 (IRS-1) is detectable in follicles extracted from typical human ovaries, with its expression intensity rising alongside follicular growth [74]. Additionally, within granulosa cells derived from normal ovaries, insulin enhances FSH-induced aromatase activity and collaborates with LH to promote the expression of sterol-regulatory genes encoding the LDL receptor, stAR protein, 3-hydroxysteroid-dehydrogenase (3-HSD) and cytochrome P450 side-chain cleavage (P450scc) [75]. These results imply that insulin might influence follicle growth by, at least in part, elevating the protein-level expression of its receptor substrates [76]. Moreover, insulin boosts LH receptor expression within granulosa cells obtained from developing human follicles [77].

IR could play its detrimental role on ovarian function by impairing the aforementioned pathways. Granulosa cells cultured from PCOS anovulatory patients show resistance to insulin effects on glucose metabolism compared to cells from healthy women [78]. However, these cells maintain normal steroidogenic output when exposed to physiological insulin doses. Furthermore, follicles retrieved from polycystic ovaries show abnormal patterns of expression of IRS-1 and -2 compared to healthy ones [76]. In particular, theca and granulosa cells exhibited elevated IRS-2 staining intensity while concurrently displaying decreased IRS-1 staining only in granulosa cells. These findings suggest that IR could specifically impact carbohydrate metabolism by selectively reducing IRS-1 in granulosa cells, leading to a compensatory rise in IRS-2 expression. Additionally, given the recognized antiapoptotic role of IRS-2, its increased presence in small antral follicles could indicate a potential association with cyst accumulation in polycystic ovaries [76].

The accumulation of visceral fat in the context of obesity sustains inflammatory stress and IR within adipose tissue through the ongoing interaction between macrophages and adipocytes. Essential mediators in this process are inflammatory cytokines, collectively referred to as adipokines, which are generated by either adipocytes or macrophages in adipose tissue [58,79]. In particular, cytokines secreted by macrophages hinder insulin activity in cultured adipocytes [80]. This inhibition is achieved through the downregulation of GLUT4 and IRS-1, a reduction in Akt phosphorylation and the impairment of insulin-stimulated GLUT4 translocation [81]. Adiponectin, the predominant adipokine primarily released by cells in visceral adipocytes, exhibits diminished levels in both lean and obese women diagnosed with PCOS, in comparison to controls with matched BMIs [82,83]. A separate category of compounds, advanced glycated end products (AGEs), recognized for their involvement in inflammatory and oxidative pathways, was observed to be notably higher in women diagnosed with PCOS [84].

The most commonly utilized therapeutic strategies for PCOS are based on the molecular mechanisms mentioned above. A recent meta-analysis has shown that the use of metformin significantly increases serum adiponectin levels while simultaneously reducing tumor necrosis factor-α (TNF-α) and C-reactive protein (CRP) concentrations [85].

Additionally, nutraceuticals, along with physical exercise and a balanced diet, may significantly improve the metabolic milieu of patients with PCOS. In particular, the appropriate use of older-generation molecules such as inositols [86] or the incorporation of modified resveratrol into the therapeutic regimen [87] could bring benefits to the ovarian environment, boosting the follicular metabolism and protecting cells from oxidative stress.

### 3.4. Metformin: A Final Solution for All Patients with PCOS?

Metformin is a biguanide that has been used to treat type 2 diabetes for almost seven decades. In recent decades, it has also been used in the treatment of PCOS, albeit off-label [88]. Metformin has pleiotropic effects, and its mechanism of action is based on the activation of 5’-AMP-activated protein kinase (AMPK), an enzyme involved in insulin signaling and fat and glucose metabolism. This results in reduced expression of genes for enzymes involved in gluconeogenesis and lipogenesis and thus reduced hepatic gluconeogenesis and lipogenesis [89]. In addition, it affects the binding of insulin to the receptors and thus increases the uptake of glucose into the cells of peripheral tissues, reduces the activity of acetyl-coenzyme A carboxylase, which is involved in the synthesis of fatty acids, and increases the oxidation of free fatty acids [89].

It has been shown that taking metformin in PCOS patients reduces the extent of hyperinsulinemia. This is closely related to an increase in the concentration of IGFBP-1 and a decrease in IGF-1. It is important to point out that although metformin therapy leads to a decrease in insulin concentration, it does not alter the secretory profile in any way and does not cause hypoglycemia; on the contrary, it maintains euglycemia. It has been found that taking metformin reduces the progression of glucose intolerance, i.e., prediabetes, to type 2 diabetes by an average of 31%, or 48% in extremely adherent individuals [90].

Metformin also lowers serum testosterone concentrations by 20–25% [91]. It is hypothesized that the causes of such a decrease in androgen concentration are reduced LH secretion and reduced aromatase activity due to improved insulin sensitivity. Some do not rule out either a direct effect of metformin on the ovaries or an increase in tyrosine kinase activity of the insulin receptors. Despite the positive effects on hyperandrogenemia, metformin monotherapy has very little or no effect on the clinical signs of hyperandrogenism [92].

The reduction in circulating insulin concentrations leads, among other things, to a normalization of the pulsatile secretion of GnRH and gonadotropin, which indirectly triggers ovulation [93]. It is assumed that metformin not only increases the frequency of ovulation and also the pregnancy rate—but not the birth rate—in PCOS patients; therefore, metformin monotherapy is not recommended as a first-line treatment for anovulatory infertility [94]. On the other hand, clomiphene citrate proved to be more effective in terms of ovulation, achieving pregnancy and childbirth compared to metformin alone. However, in clomiphene-resistant patients, the addition of metformin improves ovulation, pregnancy and birth rates compared to clomiphene citrate monotherapy [95]. It is therefore assumed that the combination of metformin and clomiphene citrate is an effective therapy in the treatment of infertility in patients who are resistant to clomiphene citrate. This raises the question of the safety of taking metformin during pregnancy. The potential benefits of taking this drug during pregnancy are a lower risk of spontaneous abortion and the development of gestational diabetes [96]. Animal studies conducted to date have not shown any risks associated with its use during pregnancy, but due to the lack of controlled studies in pregnant women, the use of metformin in pregnant women cannot be officially recommended [97].

Due to the inconsistent results of the studies, no generally valid conclusions can be drawn about the influence of metformin on the anthropometric characteristics of PCOS patients. It is almost certain that metformin does not lead to weight gain, but its use reduces weight gain only minimally or not at all [98]. The best results are likely achieved with a combination of metformin and lifestyle changes. In these cases, a moderate loss of body weight and visceral adipose tissue and a reduced BMI and waist-to-hip ratio were observed. Among other things, metformin has a protective effect on the cardiovascular system and reduces the cardiovascular complications of type 2 diabetes as well as the mortality rate from this form of diabetes. Metformin has been found to lower systolic blood pressure, reduce triglyceride and LDL cholesterol levels and increase HDL cholesterol levels, thus ensuring a less atherogenic lipid profile [99]. Metformin is a relatively well-tolerated drug. The main side effect of this drug is lactic acidosis, although the likelihood of this occurring is generally low. Existing liver and kidney diseases favor the occurrence of these dangerous side effects; therefore, the use of metformin is contraindicated in such cases. Other side effects are mainly gastrointestinal in nature, such as diarrhea, flatulence, nausea, vomiting and a metallic taste in the mouth [100]. To reduce them, patients are advised to take metformin with their main meal and gradually increase the dose from 500 mg to 2 g per day or use the prolonged-release form. Finally, metformin is the drug of choice for the treatment of glucose metabolism abnormalities due to its good safety profile and its favorable immediate and long-term effects on the health of PCOS patients [101].

### 3.5. GLP-1 Receptor Agonists: Current Knowledge on Their Therapeutic Potential in PCOS

GLP-1 receptor agonists (GLP-1RAs) such as exenatide, liraglutide and semaglutide have emerged as new therapeutic options for PCOS due to their clear benefits in the treatment of metabolic disorders [102]. Longer-acting GLP-1RAs and once-weekly formulations have shown better potential for lowering blood glucose levels and less gastrointestinal discomfort compared to their shorter-acting counterparts [103]. Certain agonists, especially at higher doses, have also demonstrated benefits for weight loss. In people with diabetes, the use of GLP-1RAs has been associated with a significant reduction in glycosylated hemoglobin, weight loss, a modest reduction in blood pressure and an improvement in hyperlipidemia [102].

In overweight women with PCOS, exenatide improved menstrual cyclicity, ovulation rate and free androgens while improving glucose tolerance and reducing abdominal fat [102]. Furthermore, exenatide appears to be more effective in combination with metformin rather than either monotherapy [104], showing more extensive metabolite profiles as a result of the regulation of multiple metabolism pathways involved in pathogenesis of obesity in PCOS patients [102,103]. Moreover, women receiving exenatide monotherapy showed significantly higher rate of natural pregnancy with PCOS after alleviating insulin resistance [102].

A randomized study involving women with PCOS and obesity previously treated with metformin reported a greater reduction in BMI with daily administration of 1.2 mg liraglutide compared to 1000 mg metformin administered twice daily [104]. Treatment with liraglutide also significantly reduced the area of visceral adipose tissue. Subcutaneously administered liraglutide at a dose of 3 mg once daily is indicated as an adjunct to chronic weight management in adults with a BMI of ≥30 kg/m^2^ or a BMI of ≥27 kg/m^2^ and at least one weight-related comorbidity, including hypertension, dyslipidemia, obstructive sleep apnea or type 2 diabetes [105].

Two studies investigated GLP-1 secretion in average-weight women with PCOS compared to age- and BMI-matched healthy women. In the first study, the authors examined fasting and post-meal GLP-1 levels and found that both levels were significantly reduced in the women with PCOS compared to the controls [105]. This finding suggests altered dynamics of incretin secretion in PCOS, which could contribute to the risk of type 2 diabetes. Although not yet reported, it suggests that this impaired GLP-1 secretion may be even more pronounced in women with PCOS and obesity. In the second study, the authors observed that GLP-1 levels were similar in the PCOS and control subjects in the early phase of the 75 g oral glucose tolerance test (OGTT) up to 60 minutes, but that GLP-1 levels were significantly lower after 180 min, suggesting that women with PCOS may have decreased GLP-1 secretion in the late postprandial phase [106].

In addition to the effect of weight loss, several studies provided additional insights into the metabolic benefits of GLP-1RAs in PCOS. Due to the high prevalence of prediabetes in PCOS, additional insights were gained from a study examining metformin, exenatide and their combination to investigate their effects on the remission rate of prediabetes. The remission rate of the combination group or the exenatide group was significantly higher than that of the metformin group (64% and 56% vs. 32%), most likely due to the improvement in postprandial insulin secretion [100]. Importantly, the effects of exenatide therapy persisted after 12 weeks of drug discontinuation, indicating a possible cellular metabolic legacy effect of the treatment [100].

Moreover, there were two studies that looked at pregnancy rates in women with PCOS after intervention with GLP-1RAs prior to conception, and both reported better pregnancy outcomes after discontinuation of GLP-1RAs [107,108]. The first study involved 176 overweight or obese women with PCOS and looked at the natural pregnancy rates over the following 12 weeks after 12 weeks of treatment with exenatide [107]. The main outcome of the study, the natural pregnancy rate after pre-treatment, was significantly higher in the exenatide group than in the metformin group (43.6% and 18.70%, respectively). Although the study was not designed to investigate the underlying mechanisms of this difference in reproductive outcomes, the authors suggest that weight loss is most likely the main reason for the improved fertility. The second study included 28 obese women with PCOS and investigated an intervention with low-dose liraglutide (1.2 mg QD) in combination with metformin [108]. The combination of liraglutide and metformin was superior to metformin alone in increasing both in vitro fertilization and cumulative (including spontaneous conception) pregnancy rates after pretreatment in patients who were previously resistant to reproductive treatment. The pregnancy rate per embryo transfer was 85.7% in the combination group compared to 28.6% in the metformin alone group.

## 4. Discussion

### Key Findings and Future Perspectives

The existing research on GLP-1 receptor agonists (GLP-1RAs) has not conclusively established a specific therapeutic use for these drugs. There is growing interest in GLP-1RAs as potential treatments for obesity-related issues, including metabolic irregularities, hyperandrogenism, infertility and menstrual disruptions, particularly in women with polycystic ovary syndrome (PCOS) [106]. Nevertheless, it remains unclear whether the positive effects of GLP-1RAs in PCOS are solely attributable to weight loss and metabolic improvements, or if they also directly impact the hypothalamic–pituitary–gonadal (HPG) axis in PCOS [105,106].

Recently, a metanalysis of eight randomized controlled trials (RCTs) was performed. The findings indicated that GLP-1RAs, used alone or alongside metformin, were significantly more effective in reducing body weight, waist circumference and body mass index compared to metformin alone. A further sensitivity analysis reinforced the stability and reliability of these meta-analysis results. Indeed, the anti-obesity effects of GLP-1RAs, either alone or in combination with metformin, were more effective than metformin alone in achieving weight loss, reducing waist circumference, and lowering BMI in overweight or obese PCOS patients [109].

A notable limitation in current RCTs is their focus predominantly on PCOS patients with severe obesity. This demographic does not fully represent the broader spectrum of PCOS cases commonly seen in clinical settings. Moreover, most of these studies have administered GLP-1RAs at low to medium doses. Given that certain effects, especially weight reduction, are known to be dose-dependent and more significant at higher dosages, the potential benefits of GLP-1RAs on the clinical and metabolic aspects of PCOS may not be fully observable in the existing studies. This indicates a need for further RCTs to explore the impacts of higher doses of GLP-1RAs. Additionally, the efficacy of the newer generation of GLP-1RAs, particularly in terms of dosage and duration of exposure, warrants more extensive research.

For example, GLP-1RAs have shown promising cardiovascular benefits. A recent study investigated the effects of GLP-1RAs on the proliferation and migration of vascular smooth muscle cells (VSMCs) exposed to angiotensin II (Ang II) and high concentrations of inorganic phosphate (Pi). The treatments used were exendin-4, liraglutide and dulaglutide. The results indicated that Ang II notably increased VSMC proliferation and migration, along with upregulating gene and protein expression linked to cell proliferation [104]. However, the GLP-1RAs significantly reduced these effects, decreasing gene and protein expression associated with proliferation. The use of specific inhibitors showed that these reductions were mediated through pathways involving extracellular signal-regulated kinase (Erk) and c-JUN N-terminal kinase (JNK). The inhibition of GLP-1 receptor through siRNA reversed these reductions. Additionally, GLP-1 (9–36) amide also lessened the proliferation and migration in Ang II-treated VSMCs. Furthermore, GLP-1RAs decreased calcium deposition in Pi-treated VSMCs by inhibiting the activating transcription factor 4 (Atf4). In conclusion, GLP-1RAs effectively mitigate abnormal VSMC proliferation and migration through both GLP-1 receptor-dependent and independent pathways and also inhibit vascular calcification induced by Pi [110].

Moreover, GLP-1RAs have been found to have anti-inflammatory properties that are beneficial in managing the chronic complications associated with type 2 diabetes. It is known that GLP-1RAs reduce T cell-driven inflammation in the gut systemically by directly targeting the GLP-1 receptors on gut intraepithelial lymphocytes [110]. However, the mechanism through which they reduce systemic inflammation in the absence of extensive immune system expression of GLP-1R relies on the fact that activating GLP-1R curtails the production of plasma tumor necrosis factor alpha (TNF-α) in response to various Toll-like receptor agonists. Interestingly, these effects are not brought about by GLP-1Rs located in hematopoietic or endothelial cells, but rather are dependent on central neuronal GLP-1Rs. In particular, GLP-1R activation decreases TNF-α levels through α1-adrenergic, δ-opioid and κ-opioid receptor signaling pathways. These findings contribute to the growing understanding of brain–immune system interactions and suggest a novel gut–brain GLP-1R axis as a mechanism for the suppression of peripheral inflammation [111].

In preclinical and randomized human studies, semaglutide rose as potent agent for PCOS management, with roles in the pathogenesis of PCOS as well as food metabolism regarding obesity management. Animal study by Zhang and colleagues showed the protective role of liraglutide and semaglutide in treating PCOS in mice, with improvements in glucose metabolism, metabolic syndrome and sex hormone abnormalities in mice sera. Furthermore, reduced levels of insulinemia and androgenemia were followed by decreased inflammatory mediator levels (TLR-4 and NFkB) and stimulating brown adipose tissue activity [112]. Subcutaneous administration of semaglutide (1.0 mg) also showed 4 h delayed gastric emptying of solid foods in a placebo-controlled trial of obese women with PCOS compared to long-acting liraglutide with a 1-hour gastric delay. Furthermore, the study showed significant decreases in androgens in the semaglutide group [113]. The exact dosage of semaglutide reduced tongue fat storage in women with PCOS, with studies being conducted to investigate semaglutide effects on taste perception [114].

Despite promising positive effects presented in recent studies, there are still no recommended therapy regimens employing GLP agents in terms of dosage and intervention duration for the management of obesity and/or PCOS. GLP agents are administered subcutaneously via prefilled pens [102]. Semaglutide has been shown to provide effects in intervals of 12–16 weeks, during which time it was subcutaneously administered once weekly in an incremental dosage regime of 0.25 mg or 0.5 mg during the first 4 weeks up to 1.0 mg in the remaining weeks [113,114]. An exenatide intervention used in recent studies was administered in intervals of 12 up to 24 weeks in different regimens. The patients were receiving initial dose of 5 ηg twice daily in the first 4 weeks, increasing up to 10 ηg twice daily for 20 weeks. Another regime included 10ηg of exenatide twice daily and 2 mg once weekly for 12 weeks [113,114,115]. Liraglutide was administered daily, as previously mentioned, with 1.2 mg up to 3.0 mg once daily [105]. Understanding the optimal dosing and treatment duration could significantly enhance the therapeutic use of GLP-1RAs in managing PCOS and its associated conditions. This expansion in knowledge is crucial to fully grasp the potential of GLP-1RAs in treating the multifaceted symptoms of PCOS.

## Figures and Tables

**Figure 1 biomedicines-12-00653-f001:**
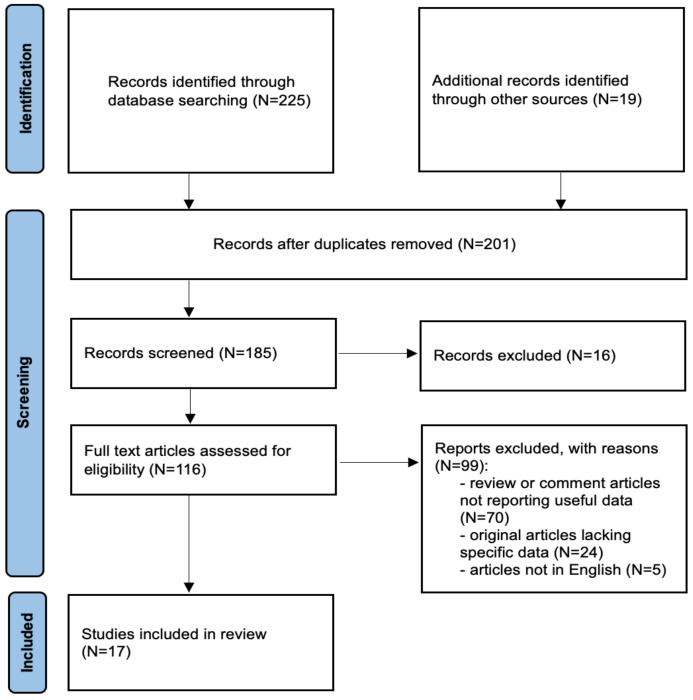
Flow diagram of narrative review search.

**Table 1 biomedicines-12-00653-t001:** PCOS phenotypes according to The National Institutes of Health consensus conference panel [17].

Phenotypes	Hyperandrogenism	Ovulatory Dysfunction	Polycystic Ovarian Morphology
A	+	+	+
B	+	+	-
C	+	-	+
D	-	+	+

**Table 2 biomedicines-12-00653-t002:** Aberrant physiological pathways involved in the pathogenesis of chronic anovulation [16,18].

Factor	Abnormality in PCOS	Result
FSH	Decreased secretion → relative deficiency	Inadequate follicular stimulation
LH	Hypersecretion	Hyperandrogenemia, follicle growth interruption
Insulin	Hypersecretion → decreased glucose tolerance → insulin resistance	Hyperandrogenemia, follicle growth interruption
Androgens	Hypersecretion	Abnormal gonadotropin secretion, follicle growth interruption
Estrogens	Hypersecretion	Suppression of FSH secretion, increased LH secretion
Inhibin B	Hypersecretion	Suppression of FSH secretion
Apoptosis	Reduced	Increased number of small follicles included in steroidogenesis
Growth factors	Abnormal expression	Abnormal apoptosis (EGF/TGF-α), follicle growth interruption (TGF-β), increase secretion of granulosa lutein cells (VEGF), suppression of estrogen synthesis (IGF-1), excessive androgen production (IGFBP-1)

Abbreviations: EGF—endothelial growth factor; IGFBP-1—insulin-like growth factor binding protein 1; IGF-1—insulin-like growth factor 1; TGF-α—transforming growth factor-α; TGF-β—transforming growth factor-β; VEGF—vascular endothelial growth factor.

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
