# Peer review of "Incretin Hormone Secretion in Women with Polycystic Ovary Syndrome: Roles of Obesity, Insulin Sensitivity and Treatment with Metformin and GLP-1s"

_biomedicines, 2024, doi:10.3390/biomedicines12030653_

Round 1

Reviewer 1 Report

Comments and Suggestions for Authors

The narrative review aims to delineate the mechanisms underlying infertility

and polycystic ovary syndrome (PCOS), specifically emphasizing the

influence of obesity, insulin sensitivity, and the therapeutic implications of

metformin and GLP-1s. However, several issues require attention.

1. The author revealed that there’s already a meta-analysis for the utilization

of GLP-1RAs, and pointed out the importance of dosage and duration of

exposure; however, the presented results did not specify this related

information.

2. The materials and methods are relatively rough, and how the author

narrowed down the treatment and the results were not mentioned. How

the included 17 studies weighted the results was not clear to me.

3. Table 1 and 2 should have added related references.

4. The narrative review should address the aspect of innovation more

explicitly, particularly in distinguishing between well-established

knowledge and novel insights. Some information presented may already

be well-known within the field, but the review should strive to identify any

novel findings, perspectives, or approaches that contribute to advancing

understanding in the field of infertility and PCOS.

Author Response

1. Summary

Thank you very much for taking the time to review this manuscript. Please find the detailed responses below and the corresponding revisions/corrections in track changes in the re-submitted files.

Comments 1: The narrative review aims to delineate the mechanisms underlying infertility and polycystic ovary syndrome (PCOS), specifically emphasizing the influence of obesity, insulin sensitivity, and the therapeutic implications of metformin and GLP-1s.

However, several issues require attention.

The author revealed that there’s already a meta-analysis for the utilization of GLP-1RAs, and pointed out the importance of dosage and duration of exposure; however, the presented results did not specify this related information.

Response 1: First of all, thank you for your precious suggestions and favorable comments. Regarding importance of dosage and duration of exposure, we agree with your comments. Studies which have been included in mentioned meta-analysis have limitations regarding the dose administered (‘’A notable limitation in current RCTs is their focus predominantly on PCOS patients with severe obesity. This demographic does not fully represent the broader spectrum of PCOS cases commonly seen in clinical settings. Moreover, most of these studies have administered GLP-1RAs at low to medium doses. Given that certain effects, especially weight reduction, are known to be dose-dependent and more significant at higher dosages, the potential benefits of GLP-1RAs on the clinical and metabolic aspects of PCOS may not be fully observable in the existing studies.’’)

Comments 2: The materials and methods are relatively rough, and how the author narrowed down the treatment and the results were not mentioned. How the included 17 studies weighted the results was not clear to me.

Response 2: We agree with your comment and we have modified Materials and Methods section with PICO algorithm – (‘’This study aimed to ask the following PICO questions:

Population: women with verified PCOS according to the relevant criteria.

Intervention: metformin and GLP1 receptor agonists assessment.

Comparison: metformin and GLP1 receptor agonists comparison and its role in insuline resistance and obesity.

Outcome: therapeutic efficacy of metformin and GLP1 receptor agonists.’’)

Comments 3: Table 1 and 2 should have added related references.

Response 3: Thank you for your precious suggestions – we have added related references for both tables.

Comments 4: The narrative review should address the aspect of innovation more explicitly, particularly in distinguishing between well-established knowledge and novel insights. Some information presented may already be well-known within the field, but the review should strive to identify any novel findings, perspectives, or approaches that contribute to advancing understanding in the field of infertility and PCOS.

Response 4: Thank you for your comments – we have inserted a very recent study (Jiang, N.-X.; Zhao, W.-J.; Shen, H.-R.; Du, D.; Li, X.-L. Hyperinsulinemia impairs decidualization via Akt-NR4A1 signaling: New insight into polycystic ovary syndrome (pcos)-related infertility. Journal of Ovarian Research 202417.) from February 2024 which proposed very interesting and novel mechanism which connects hyperinsulinemia and infertility in PCOS population (‘’The most recent publications have revealed molecular mechanisms which are responsible for reproductive issues in this particular population. For instance, a study by Jiang and associates demonstrated hyperinsulinemia as the main factor in PI3K/AKT-NR4A pathway disruption which leads to decidualization defects and endometrial dysfunction, thereby contributing to infertility in PCOS patients’’).

Reviewer 2 Report

Comments and Suggestions for Authors

The manuscript “`Incretin hormone secretion in women with polycystic ovary syndrome: roles of obesity, insulin sensitivity and treatment with metformin and GLP-1s” provides good information on how metformin and GLP-1s have affected polycystic ovary syndrome (POC).

Personally, I believe that the accumulation of results from research like this manuscript is highly likely to provide new diagnostic methods for infertility treatment in the future.

To realize the applicable possibility of this diagnostic method in detail in the future, the authors need to discuss the methods for collecting data.

Some points have to be corrected.

Major points

1. It is necessary to classify and explain the growth factors in Table 1 in more detail. Growth factors include IGFs, TGF-beta family, and so on.

2. If it is to be applied clinically, it is necessary to discuss in the discussion which timing should be used to increase the chance of success.

Comments on the Quality of English Language

Minor editing of English language required.

Author Response

1. Summary

Thank you very much for taking the time to review this manuscript. Please find the detailed responses below and the corresponding revisions/corrections in track changes in the re-submitted files.

Comments 1: The manuscript “Incretin hormone secretion in women with polycystic ovary syndrome: roles of obesity, insulin sensitivity and treatment with metformin and GLP-1s” provides good information on how metformin and GLP-1s have affected polycystic ovary syndrome (POC). Personally, I believe that the accumulation of results from research like this manuscript is highly likely to provide new diagnostic methods for infertility treatment in the future.

To realize the applicable possibility of this diagnostic method in detail in the future, the authors need to discuss the methods for collecting data.

Some points have to be corrected.

Major points

It is necessary to classify and explain the growth factors in Table 2 in more detail. Growth factors include IGFs, TGF-beta family, and so on.

Response 1: First of all, thank you for your precious suggestions and favorable comments. We have made necessary corrections in Table 2, incorporating specific role of the most important growth factors in PCOS pathophysiology. We have also added abbreviations below the table for better understanding.

Comments 2: If it is to be applied clinically, it is necessary to discuss in the discussion which timing should be used to increase the chance of success.

Response 2: We agree with your comment and we have modified Discussion section regarding potential treatment duration. Your comment is very insightful and, therefore, should provide further investigation in this particular field.

Reviewer 3 Report

Comments and Suggestions for Authors

This is a narrative review aiming to present the mechanisms linking obesity and insulin resistance with infertility in women with PCOS and the implications and consequences of metformin and GLP-1 agonists treatment.

The subject is quite vast, and I believe the most interesting and original part should be the review of the efficacity of metformin and GLP1 receptor agonist treatment on different aspects of PCOS pathology. However, if the section dedicated to metformin is extensive and updated, this is definitely not the case when GLP-1 R agonists are presented. In my opinion, this should be the highlight of the review, but its is too short, it does not give enough information regarding the existent research on this topic. For example, regarding liraglutide treatment in patients with PCOS, the authors only mention one study -using low dose liraglutide. A brief literature search would show the existence of several studies using full dose (3 mg) liraglutide, with different outcomes – weight loss, decreasing androgen levels, reducing insulin resistance, etc. Also, there are some studies involving newer products, such as semaglutide – these are also not mentioned in this dedicated review

In conclusion, my opinion is that this might be an interesting review, provided that extensive changes and additions are performed in the GLP-1 R agonist chapter – more evidence-based data,  some insights regarding ongoing trials and potential new “players” in this field

Author Response

1. Summary

Thank you very much for taking the time to review this manuscript. Please find the detailed responses below and the corresponding revisions/corrections in track changes in the re-submitted files.

Comments 1: This is a narrative review aiming to present the mechanisms linking obesity and insulin resistance with infertility in women with PCOS and the implications and consequences of metformin and GLP-1 agonists treatment.

The subject is quite vast, and I believe the most interesting and original part should be the review of the efficacity of metformin and GLP1 receptor agonist treatment on different aspects of PCOS pathology. However, if the section dedicated to metformin is extensive and updated, this is definitely not the case when GLP-1 R agonists are presented. In my opinion, this should be the highlight of the review, but its is too short, it does not give enough information regarding the existent research on this topic. For example, regarding liraglutide treatment in patients with PCOS, the authors only mention one study -using low dose liraglutide. A brief literature search would show the existence of several studies using full dose (3 mg) liraglutide, with different outcomes – weight loss, decreasing androgen levels, reducing insulin resistance, etc. Also, there are some studies involving newer products, such as semaglutide – these are also not mentioned in this dedicated review.

In conclusion, my opinion is that this might be an interesting review, provided that extensive changes and additions are performed in the GLP-1 R agonist chapter – more evidence-based data,  some insights regarding ongoing trials and potential new “players” in this field.

Response 1: First of all, thank you for your precious suggestions and favorable comments. We have made necessary corrections and updates according to your advice in Discussion section – please see the track changes in the manuscript. Although there are lack of data regarding PCOS population, we have added recent knowledge about potential new ‘’players’’ in the field.

Round 2

Reviewer 1 Report

Comments and Suggestions for Authors

The author improved manuscript well and can be accepted.

Author Response

1. Summary

Thank you very much for taking the time to review this manuscript. Please find the detailed responses below and the corresponding revisions/corrections in track changes in the re-submitted files.

Comments 1: The author improved manuscript well and can be accepted.

Response 1: Thank you for your encouraging comments.

Reviewer 2 Report

Comments and Suggestions for Authors

I think that the revised manuscript has been improved.

Author Response

1. Summary

Thank you very much for taking the time to review this manuscript. Please find the detailed responses below and the corresponding revisions/corrections in track changes in the re-submitted files.

Comments 1: I think that the revised manuscript has been improved.

Response 1: Thank you for your encouraging comments.

Reviewer 3 Report

Comments and Suggestions for Authors

Some of my previous comments still apply and were not adequately adressed by the authors:

-       The patophysiological considerations are too long

-       GLP-1 receptor agonists are presented briefly, not focusing on the published studies describing their impact on different clinical components of PCOS

Author Response

1. Summary

Thank you very much for taking the time to review this manuscript. Please find the detailed responses below and the corresponding revisions/corrections in track changes in the re-submitted files.

Comments 1: Some of my previous comments still apply and were not adequately adressed by the authors:

The patophysiological considerations are too long.

Response 1: Thank you for your insightful comment. We have modified the section 3.1. PCOS: Definition and Current Pathophysiological Considerations according to your suggestion as we have removed the redundant text.

Comments 2: GLP-1 receptor agonists are presented briefly, not focusing on the published studies describing their impact on different clinical components of PCOS.

Response 2: Thank you for your guidance to improve manuscript. We have add necessary text in section 3.5. GLP-1 receptor agonists: Current Knowledge on Their Therapeutic Potential in PCOS. We have included current knowledge about pregnancy rate and other metabolic benefits previously not mentioned.

Round 3

Reviewer 3 Report

Comments and Suggestions for Authors

Manuscript was improved.